# Overcoming the Pitfalls of Vision-Language Model for Image-Text Retrieval

## ABSTRACT

This work tackles the persistent challenge of image-text retrieval, a key problem at the intersection of computer vision and natural language processing. Despite significant advancements facilitated by large-scale Contrastive Language-Image Pretraining (CLIP) models, we found that existing methods fall short in bridging the fine-grained semantic gap between visual and textual representations, particularly in capturing the nuanced interplay of local visual details and the textual descriptions. To address the above pitfalls, we propose a general framework called Local and Generative-driven Modality Gap Correction (LG-MGC), which devotes to simultaneously enhancing representation learning and alleviating the modality gap in cross-modal retrieval. Specifically, the proposed model consists of two main components: a local-driven semantic completion module, which complements specific local context information that overlooked by traditional models within global features, and a generative-driven semantic translation module, which leverages generated features as a bridge to mitigate the modality gap. This framework not only tackles the granularity of semantic correspondence and improves the performance of existing methods without requiring additional trainable parameters, but is also designed to be plug-and-play, allowing for easy integration into existing retrieval models without altering their architectures. Extensive qualitative and quantitative experiments demonstrate the effectiveness of LG-MGC by achieving consistent state-of-the-art performance over strong baselines. *The code is included in the supplementary material.*

## 1 INTRODUCTION

Image and text are two pivotal information carriers to help human and intelligent agents to better understand the real world. Numerous studies [20, 28, 51, 55] have been undertaken in both the fields of computer vision and natural language processing to bridge these two modalities. As a fundamental yet intricate topic, image-text retrieval benefits a variety of applications such as person search, sketch-based image retrieval, and food recipe retrieval, to name but a few [2, 19, 25, 42, 46, 59].

Although Image-text retrieval has garnered significant attention in recent years [15, 20, 55], the fundamental challenges, such as accurately and efficiently learning cross-modal embeddings and bridging the inter-modality gap between images and texts, are far from being resolved. *The former challenge stems from the complex visual appearances of images, contrasted with the abstract semantics of texts.* Specifically, characterized by rich details and contextual scene information, it is challenging to effectively capture and distill the visual information into a meaningful and discriminative representation. Furthermore, textual data often represent the same visual concept in abstract and variable ways. This misalignment necessitates sophisticated feature extraction and representation learning techniques to effectively capture these nuances. *The latter challenge, intrinsic to cross-modal tasks, arises from inherent representation disparities between vision and language.* Vision-based models typically

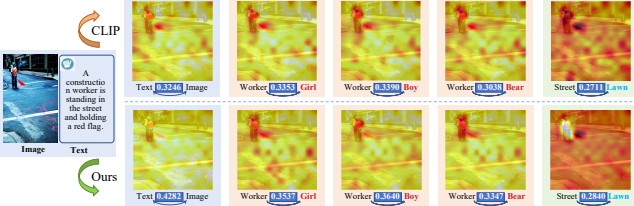

**Figure 1: Similarity maps from the vanilla CLIP and our proposed model on Flickr30K test set. From left to right, the maps illustrates the smooth patch-wise similarity between the image patches and the original text, along with four modified texts (i.e., worker→{girl, boy, bear} and street→lawn). The brighter the color, the higher the value.**

process inputs as continuous and multi-dimensional arrays of pixel values, making the visual information fundamentally different from the discrete textual data. Bridging this heterogeneous gap requires not only mapping different modality data to a common feature space but also doing so in a way that aligns semantic related but representation distinct entities.

In addressing the challenges, prevailing image-text retrieval methods can be classified into two paradigms. The first, known as the *score-based* matching approach [3, 6, 27], involves cross-modal interaction between local visual and textual features to obtain a cumulative similarity score. However, due to the large depth of interaction between modalities, such methods significantly lag in processing speed, making them suboptimal for large-scale cross-modal retrieval. In response, the *embedding-based* matching approach, which employs a dual-encoder architecture, has become increasingly favored for its efficiency in facilitating retrieval [20, 34, 47]. This approach first employs two dedicated encoders to generate features for images and texts separately, and then cultivates a joint image-text embedding space by constraining the coarse-grained alignment between global image and text features. Nevertheless, this coarse-grained alignment constraint tends to overlook the intricate semantics of images and texts, ultimately constraining the image-text matching performance.

Fortunately, recent works using the large-scale Contrastive Language-Image Pretraining (CLIP) model [4] have shown great potential in improving the performance of the embedding-based methods through learning robust features (addressing the first challenge). However, studies [1, 38] also find that although the performance of the cross-modal retrieval task is greatly improved, it is still challenging to learn specific fine-grained visual and textual concepts. Furthermore, we found that the CLIP model [4] tends to be biased toward dominant pixels, which causes micro-objects containing critical information to be represented at greater distances from the textual representation than the background. To validate this conclusion, we calculate the similarity between a given image and the corresponding original text description, as well as with texts where key information within the image have been altered, such

as replacing 'worker' with 'girl' , 'boy', and 'bear'. The similarity map with the score is shown in Fig. 1. From the results in the first four columns (in the first row) of the figure, it is evident that CLIP is insensitive to local changes, because the similarity between the image and its altered text descriptions aligns with, or is even higher than (when 'worker' is replaced with 'girl' and 'boy'), the similarity with the original text. However, replacing words in the text that describe background elements, such as substituting 'street' with 'lawn', leads to a noticeable decline in the similarity score, dropping from 0.3246 to 0.2711. Our analysis of the underlying causes behind this phenomenon highlights that the CLIP model, trained via contrastive loss, aims to match an image with the text based on global features derived from the class token, without explicitly capturing local features. Although it incorporates numerous self-attention modules to enhance information exchange among different image patches, it tends to overlook patches that do not contain dominant pixels, such as the object 'worker'. Yet, this oversight is crucial for fine-grained cross-modal retrieval, particularly in the real-world scenarios. Therefore, to overcome this pitfall, making the retrieval model sensitive to both local and global information is worthy of exploration. In this paper, we propose a straightforward yet potent approach for the simultaneous explicit and implicit integration of local information into the global representation.

Furthermore, in addressing the second challenge, namely the heterogeneous modality gap, existing CLIP models primarily rely on contrastive loss to narrow the distance between paired data and widen the gap between unpaired data. In this way, it can map the image and text into a shared representation space. However, recent study [32] suggests that the inherent inductive bias of deep neural architectures leads to a phenomenon known as the 'cone effect', and different encoders will create different cones, which exacerbates the modality gap in image-text retrieval. Additionally, it also shows that relying solely on the naive global-focused contrastive learning that frequently employed by multi-modal models fails to adequately bridge this gap. In this case, a common solution is to translate data from one modality to another, and then measure the similarities in the transformed space. Existing methods often involve training an additional mapping model to project data from one modality to another. However, fine-tuning the over-parameterized CLIP has posed significant challenges. Increasing the model's training parameters would not only raises the training time and cost, but it may also exacerbates the risk of overfitting and produces unsatisfactory results. Thus, to address this pitfall, mapping data from different modalities to the same space without introducing extra parameters is of great importance. Recently, diffusion-based text-to-image generation methods have seen significant advancement and been adopted in various applications, such as semantic segmentation [37, 48], image captioning [33], and video anomaly detection [50]. This inspires us to resort to the diffusion model to mitigate the heterogeneous modality gap between image and text.

Inspired by above discussions, we propose a general Local and Generative-driven Modality Gap Correction (LG-MGC) mechanism to capture fine-grained semantics and alleviate the heterogeneous modality gap for image-text retrieval, which is designed in a plug-and-play manner for ease of integration. Specifically, the LG-MGC comprises two main modules: the first is a Local-driven Semantic Completion (LSC) module, designed to integrate effective local

information into the global representation in both explicit and implicit manners. Consequently, more comprehensive visual and textual representations can be learned (addressing challenge 1). To mitigate the modality gap in the image-text retrieval, we further develop a Generative-driven Semantic Translation (GST) module based on a fixed diffusion model. This module is responsible for the transmission of global semantics, ensuring that the overall semantic flow can be effectively transferred and aligned across different modalities, thereby narrowing the modality gap (addressing challenge 2). Through clever collaboration between the LSC and GST, as shown in the bottom row in Fig.1, our proposed model significantly enhances the performance of existing cross-modal retrieval models. Notably, it achieves these improvements without introducing additional trainable parameters, paving the way for a more intuitive and effective retrieval process.

The main contributions of this paper are three-fold. (1) We intuitively unveil the pitfalls of embedding-based image-text retrieval approaches and propose a model-agnostic method named Local and Generative driven Modality Gap Correction (LG-MGC), which serves as a plug-and-play module to enhance dual-encoder vision-language frameworks for image-text retrieval. (2) By introducing two semantic enhancement techniques, Local-driven Semantic Completion (LSC) and Generative-driven Semantic Translation (GST), we can effectively capture fine-grained cross-modal information and mitigate the heterogeneous modality gap, without adding additional trainable parameters to the baseline image-text retrieval model. (3) Extensive qualitative and quantitative experiments have demonstrated the effectiveness of our LG-MGC, achieving significant performance gains over the original CLIP and other state-of-the-art methods across two typical benchmarks.

## 2 RELATED WORK
### 2.1 Image-Text Retrieval

Image-Text Retrieval is a typical cross-modal task, whose main challenge is to learn a shared representation of images and texts and accurately measure their similarity [7, 16, 20]. According to how the cross-modal interaction is implemented, image-text matching methods can be divided into two categories, i.e., *score-based* and *embedding-based* matching. Specifically, in the domain of *score-based* approaches, fine-grained cross-modal interactions and semantic alignments occur between local fragments, followed by the computation of a cumulative similarity score [3, 6, 11, 27, 56]. For instance, SCAN [27] introduces a cross-modal attention mechanism to calculate the similarity between words and local areas of an image, facilitating local semantic alignment. IMRAM [3] delineates an iterative network designed to enhance multiple stages cross-modal interaction. NAAF [56] employs dual matching mechanisms to evaluate both similarity and dissimilarity degrees, thereby enabling a comprehensive inference of the overall similarity. In spite of the effectiveness of these methods, their efficiency is compromised mainly due to reliance on mechanisms such as cross-modal attention, iterative matching, and graph-based relationship reasoning, making them challenging to apply in large-scale cross-modal retrieval tasks. In the *embedding-based* matching methods, there typically contains a text encoder and an image encoder. The images and texts are encoded independently into a unified embedding space, with semantic similarity assessed through cosine

similarity [4, 8, 14, 20, 34, 47]. For example, VSE++ [14] designs a two-stream global feature learning network for fast image-text matching. GPO [4] proposes a learnable pooling operation to project local features into the global embedding. Benefiting the simple calculation method, the *embedding-based* retrieval model usually has a fast retrieval speed. However, due to the limited interaction between images and texts, these models primarily focus on holistic information during training and align image and text through contrastive learning, struggling to capture fine-grained cross-modal knowledge. Thereby, they face challenges in coping with the intricate heterogeneous modality gap, resulting in lower performance compared to the *score-based* methods. Our method leverages a local-driven semantic completion module learn fine-grained visual and textual representations. Furthermore, a fixed diffusion model is utilized to enhance the *embedding-based* methods through directly translating textual semantics into the visual domain. By harnessing external knowledge provided by the pre-trained visual-language model, our approach not only ensures fast retrieval speeds but also significantly enhances the model's retrieval performance.

## 2.2 Visual Language Pre-training

Vision Language Pre-training (VLP) aims to learn semantic correspondence between vision and language modalities by pre-training on large-scale image-text pairs. Inspired by the success of Transformer based [45] pre-training language model (such as BERT [10] and Vision Transformer (ViT) [12], Vision-Language Pre-training (VLP) has emerged as the prevailing paradigm in learning multimodal representations, demonstrating strong results on downstream tasks such as image captioning [9, 33, 40], cross-modal retrieval [18, 29, 58], and visual question answering [17, 24, 44]. Most of these approaches utilize transformer based architectures, which can be categorized as *single-stream* and *dual-stream* pre-training, depending on their model structure. Specifically, in the *single-stream* models [21, 26, 49, 54, 57], text and visual features are concatenated and then fed into a single transformer encoder. Although this architecture consistently achieves high accuracy in downstream tasks, it exhibits slow retrieval speeds during the inference stage. This slowdown occurs because it needs to predict the similarity score for all possible image-text pairs, making it impractical for large-scale cross-modal retrieval tasks. Instead, dual-stream models [13, 22, 38] use two separate encoders to extract the textual and visual features independently, and these two transformer encoders do not share parameters, making it possible to calculate similarities of image-text pairs in the linear time complexity. Typical work, such as CLIP [4], exploits cross-modal contrastive pre-training by encoding image and text separately. This method allows image and text features to be computed offline, enabling efficient calculation of similarities between large-scale image-text pairs. Although this technique significantly improves the performance of cross-modal retrieval tasks by million-scale image-text contrastive pre-training, as discussed in Sec. 1 (Introduction), the dual-stream method remains challenging and ineffective for learning specific fine-grained concepts, especially when the objects do not occupy dominant pixels in the image. By contrast, our method could incorporate local information into the global visual and textual representations in both explicit and implicit manners. Furthermore, it can obtain fine-grained cross-modal information by the *dual-stream* models without introducing additional trainable parameters into them.

## 3 PROPOSED METHOD

The overall architecture of our proposed Local and Generative-driven Modality Gap Correction (LG-MGC) model is structured as a dual-encoder framework as illustrated in Fig. 2, which makes it practical for large-scale cross-modal retrieval tasks. This architecture utilizes separate comprehensive transformer-based unimodal encoders to encode the image and text before the computation of cross-modal contrastive losses. Meanwhile, we further design two pivotal modules to strengthen the semantic representation learning and cross-modal alignment, i.e., the Local-driven Semantic Completion (LSC) module, and the Generative-driven Semantic Translation (GST) module. Specifically, the LSC focuses on complementing specific local context information within global features with two criteria. The first criterion calculates the similarity between the patch (or word) token and the corresponding global embedding, and selectively injects the most dissimilar local information into the global feature to explicitly enhance the local details. The second criterion selects the features with the highest values on each channel to implicitly encode local information. As a result, our method leverages both holistic and local representation for effective image-text retrieval. Furthermore, due to the inherent inductive bias within deep neural architectures, the intrinsic modality gap challenge has never been effectively resolved. Therefore, we further propose the GST module tasked with transmitting global semantics. This module ensures that the overall semantic content is effectively transferred and aligned across various modalities. Through the strategic integration of the LSC and GST, our proposed method significantly boosts the performance of existing cross-modal retrieval models without requiring additional parameters.

**Problem formulation.** Given a set of image and text pairs, the vision and text encoder aim to encode the image $V$ and text $T$. After that, the model is required to generate a similarity score $S(t, V)$ between a text query $t \in T$ and each image based on the relevance of the textual representation and the visual feature.

## 3.1 Vanilla Image-Text Retrieval Model

Capitalizing on the renowned simplicity and substantial knowledge transfer potential of the CLIP model [38], we initialize the proposed model with the full CLIP image and text encoder to ensure it with preliminary cross-modal alignment capability, thereby establishing a solid foundational baseline.

**Image Encoder.** Following the success of vision transformer [12], the image encoder directly takes image patches as the input. By slicing an image into multiple patches, a patch sequence $V = [v_1, v_2, ..., v_n]$ is used to form a simple linear projection of pixels. To enhance the relationships among the patches, the class token [CLS] embedding is inserted into the sequence. Positional embedding is added to each patch token to encode the spatial information. The image encoder consists of a stack of $L_v$ transformer layers. Let $F_l^v$ be the input sequence of the $l$-th vision transformer layer, and then the transformation at this layer produces an output sequence that subsequently becomes the input for the next layer $(l + 1)$:

$$F_{l+1}^v = \hat{F}_l^v + MLP(LN(\hat{F}_l^v)), \quad \hat{F}_l^v = F_l^v + MHSA(LN(F_l^v)), \quad (1)$$

where $MSHA(\cdot)$ is the multi-head self-attention layer, $MLP(\cdot)$ means the multi-layer perception network, and $LN(\cdot)$ denotes the layer normalization. The input of the first transformer block is just the patch sequence $V$. Finally, the output of the last vision transformer

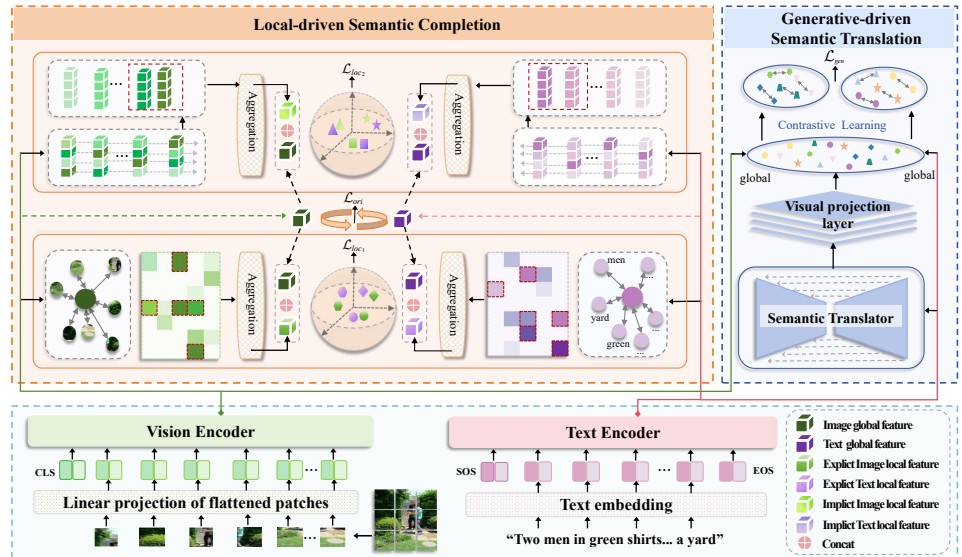

**Figure 2: The pipeline of our framework includes two key components: LSC and GST. The LSC focuses on complementing specific local information within global representations in both explicit and implicit manners. Meanwhile, the GST is dedicated to transmitting global semantics into a unified feature space to mitigate the modality gap between image and text.**

layer can be represented as $F^v = \{f^v_{cls}, f^v_1, ..., f^v_n\}$, where $f^v_{cls}$ denotes the global feature from the class token [CLS], and $f^v_i$ denotes the local feature from $i$-th patch token. Finally, the $f^v_{cls}$ is linearly projected into the image-text embedding space to obtain the ultimate global visual representation $G^v$.

**Text Encoder.** Similar to the vision encoder, we utilize the CLIP text encoder to extract the textual representation, which contains a stack of $L_t$ standard transformer layers modified by Radford et al. [38]. Following CLIP, the lower-cased byte pair encoding with a 49, 152 vocab size [43] is firstly employed to tokenize the input text description. In this way, we can convert the input text into a token sequence $T = [t_1, t_2, ..., t_m]$. Then, the text description bracketed with [SOS] and [EOS] tokens to indicate the start and end of the sequence $T$. After that, the tokenized text $T = [t_{sos}, t_1, t_2, ..., t_m, t_{eos}]$ is fed into the transformer and exploits correlations of each word by masked self-attention, resulting in a transformed textual representation $F_T = \{f^t_{sos}, f^t_1, ..., f^t_m, f^t_{eos}\}$. Finally, the [EOS] token $f^t_{eos}$ is linearly projected into the image-text embedding space to obtain the ultimate text representation $G^t$.

**Contrastive Learning.** After the feature extracting, mainstream methods for image-text retrieval typically rely on the global visual and textual feature $G^v$ and $G^t$ for similarity calculation. The model is then fine-tuned using a contrastive learning loss as follows:

$$\mathcal{L}_{ori} = \frac{1}{2}(\mathcal{L}_{v2t} + \mathcal{L}_{t2v}), \tag{2}$$

with

$$L_{v2t} = -\frac{1}{N}\sum_{i=1}^{N} log \frac{exp((G^v_i)^\top G^t_i/\tau_1)}{\sum_{j=1}^{N} exp((G^v_i)^\top G^t_j/\tau_1)},$$
$$L_{t2v} = -\frac{1}{N}\sum_{i=1}^{N} log \frac{exp((G^t_i)^\top G^v_i/\tau_1)}{\sum_{j=1}^{N} exp((G^t_i)^\top G^v_j/\tau_1)}, \tag{3}$$

where $N$ is the number of matched image-text pairs. $\tau_1$ denotes the temperature hyperparameter, which is a trainable variable. By the

contrastive loss, the model can maximize the similarity between positive image-text pairs and minimize the similarity between negative pairs, thereby realizing cross-modal retrieval.

## 3.2 Local-Driven Semantic Completion

Despite the success of previous methods, we found that relying on global embeddings to establish semantic correspondence between an entire image and a complete sentence can cause the model to overlook detailed image and text semantics. This oversight consequently impedes performance improvements in image-text retrieval. As shown in Fig. 1, we can observe that background patches in the image consistently exhibit higher similarity with the global visual representation in the vanilla CLIP model. This phenomenon indicates that the global-based image-text contrasting tends to rely heavily on dominant pixels (e.g., the background), while neglecting some significant local object information (e.g., the worker). As a result, the learned retrieval model is insensitive to the variance of objects, further making the model struggles to achieve satisfied fine-grained cross-modal retrieval performance.

Therefore, we design a local-driven semantic completion mechanism, enabling the model to simultaneously leverage global and local information during the optimization process. Instead of directly concatenating features extracted from patch and word tokens with the global feature, we meticulously develop two criteria to integrate local information into the global features, employing both explicit and implicit strategies, respectively. Specifically, in the *explicit* manner, given the encoded visual representation $F^v = \{f^v_{cls}, f^v_1, ..., f^v_n\}$ from the image encoder, we first calculate the similarity $S^v =< f^v_{cls}, \{f^v_i\}^n_{i=1} >$ between each local feature $f^v_i$ and the global feature $f^v_{cls}$, and $S^v \in R^{1\times n}$. After that, we sort the $S^v \in R^{1\times n}$ in ascending order, and opt to integrate the first $K$ local region features into the global feature $f^v_{cls}$. In other words, the $K$ local features most dissimilar to the global feature $f^v_{cls}$ are selected to enhance it. This process is straightforward and makes sense, because the local features that do not have a high similarity to

global features inevitably contain the ignored information. Through this strategy, we can explicitly integrate the overlooked local visual representations into the global feature, thereby obtaining a refined global feature $F_{exp}^v \in R^{n \times 2d}$ that is explicitly enhanced based on the selected local regions $F_{sel}^v \in R^{K \times d}$:

$$
\begin{aligned}
F_{sel}^v &= \{f_k^v\}_{k=\hat{S}_1^v}^{\hat{S}_K^v}, \quad with \ \hat{S}^v = sort_n(S^v), \\
F_{exp}^v &= Concat_d(f_{cls}^v, Mean_k(F_{sel}^v)),
\end{aligned}
\tag{4}
$$

where $sort_n(\cdot)$, $Concat_d(\cdot)$, and $Mean_k(\cdot)$ denote the similarity ranking, feature concatenation, and mean pooling operation along the dimension $n$, $d$, and $K$, respectively.

In the *implicit* manner, we automatically select the features with the greatest values on each channel. These features are expected to contain local information about important visual concepts and key entities for image-text contrasting, regardless of the foreground and background. Specifically, as shown in Fig. 2 (top), given the encoded local visual feature $F_{loc}^v = \{f_1^v, ..., f_n^v\} \in R^{n \times d}$, we first sort them in descending order along dimension $d$, and choose to incorporate the forefront $M$ responses into the global feature:

$$
\begin{aligned}
F_{loc}^{sel} &= \{[\hat{F}_{loc}^v]_m\}_{1 \le m \le M}, \quad with \ \hat{F}_{loc}^v = sort_d(F_{loc}^v), \\
F_{imp}^v &= Concat_d(f_{cls}^v, F_{loc}^{sel}).
\end{aligned}
\tag{5}
$$

Through this strategy, we can integrate the local region features into the global feature $f_{cls}^v$ with an implicit manner, and obtaining a global feature $F_{imp}^v$ that is implicitly enhanced by the local information. Similarly, we apply the same process for the textual representation $F_T = \{f_{sos}^t, f_1^t, ..., f_m^t, f_{eos}^t\}$, and obtain the explicit and implicit local-enhanced texture feature $F_{exp}^t$ and $F_{imp}^t$. Subsequently, we can calculate the *local-driven contrastive loss* $\mathcal{L}_{loc_1}$ and $\mathcal{L}_{loc_2}$ regarding with the enhanced visual feature $F_{exp}^v$, $F_{imp}^v$, and the texture feature $F_{exp}^t$, $F_{imp}^t$ by Eq. (2) and Eq. (3), respectively.

## 3.3 Generative-driven Semantic Translation

Recent studies reveal that the inherent inductive bias within deep neural architectures results in varying embedding cones across different encoders, highlighting the intrinsic challenge in the image-text retrieval, i.e., the heterogeneous modality gap. Moreover, exclusive reliance on contrastive learning has been demonstrated to be insufficient for overcoming this challenge effectively [32]. Thus, it becomes imperative to identify and develop strategies to mitigate the modality gap, with the ultimate objective of enhancing the performance of image-text retrieval. Although one modality's representation can be mapped to another modality's feature space through a complex learnable mapping network, this approach not only increases training time but also adds to the complexity of the model. Especially when data is scarce, it may lead to model overfitting and result in a decline in the retrieval performance. In fact, our ablation studies in Sec. 4.3 have also demonstrated this point. Therefore, we design a generative-driven semantic translation module based on an off-the-shelf text-to-image generation model [39], which can narrow the modality gap without increasing any trainable parameters on the vanilla CLIP model.

The objective of the GST is to directly generate the corresponding image embedding for any given text query. Subsequently, the retrieval model is optimized using both the original visual and textual representations along with the generated image embedding. *This involves two sub-problems*: how to perform cross-modal generation from text to image, and how to align the generated visual representation with the feature manifold of the original visual data. For the first problem, we propose harnessing the superior generative capabilities of the DALL·E 2 [39] to translate textual features into the visual domain. Recently, this model has demonstrated excellent performance in text-to-image generation, pushing state-of-the-art across a broad spectrum of vision and language tasks [33, 37]. Furthermore, because it consists of two-stages (i.e., a prior stage that generates CLIP image embedding given a text description, and a decoder stage that synthesizes an image conditioned on the image embedding), we can easily obtain translated textual feature under CLIP image embedding space with its prior stage. In this way, we can mitigate potential noises introduced during the image generation process and ensure a more seamless and accurate semantic translation. Technically, given a text $T$, its translated image embedding $T_{gen}$ can be calculated as follows:

$$
T_{gen} = f_\theta(T_{gen}^{(e)}, e, T), \quad with \ e \sim [1, E], \tag{6}
$$

where $f_\theta(\cdot)$ is a Gaussian diffusion model, which can generate an image embedding conditioned on the text $T$. $e$ means the iteration time, $T_{gen}^{(e)}$ denotes the generated image embedding at time $e$, and $T_{gen}^{(0)} \sim \mathcal{N}(0, 1)$ means the randomly sampled Gaussian noise.

Through the diffusion model, we can translate the textual feature into the visual domain. However, this off-the-shelf process cannot guarantee that the translated image embedding is under the unified feature distribution with the original image features, thereby imposing limitations on model optimization (i.e., the second sub-problem). Therefore, to establish connection between the generated embedding and the original visual feature, as shown in Fig. 2 (right), we introduce a projection layer $MLP_s(\cdot)$ after the diffusion model, which mirrors the architecture and shares parameters with the terminal layer of the image encoder. The transformed image embedding $\hat{T}_{gen}$ can be derived as:

$$
\hat{T}_{gen} = MLP_s(T_{gen}). \tag{7}
$$

Finally, we take the transformed image embedding $\hat{T}_{gen}$ as a bridge, and conduct contrastive learning between the $\hat{T}_{gen}$ and the global visual and textual features $G^v$ and $G^t$, respectively. Overall, the generative-driven loss can be defined as:

$$
\mathcal{L}_{gen} = \frac{1}{2}(\mathcal{L}_{g2t} + \mathcal{L}_{g2v}), \tag{8}
$$

with

$$
\begin{aligned}
L_{g2t} &= -\frac{1}{N} \sum_{i=1}^N log \frac{exp((\hat{T}_{gen}^i)^\top G_i^t / \tau_2)}{\sum_{j=1}^N exp((\hat{T}_{gen}^i)^\top G_j^t / \tau_2)}, \\
L_{g2v} &= -\frac{1}{N} \sum_{i=1}^N log \frac{exp(((\hat{T}_{gen}^i)^\top G_i^v / \tau_2)}{\sum_{j=1}^N exp((\hat{T}_{gen}^i)^\top G_j^v / \tau_2)},
\end{aligned}
\tag{9}
$$

where $\tau_2$ denotes the trainable temperature hyperparameter.

## 3.4 Training and Inference

During training, we apply a combination of the original, local-driven, and generative-driven losses to fine-tune the image and text encoder in the CLIP model for fine-grained cross-modal retrieval, formulated as follows:

$$
\mathcal{L}_{total} = \mathcal{L}_{ori} + \alpha \mathcal{L}_{loc_1} + \beta \mathcal{L}_{loc_2} + \gamma \mathcal{L}_{gen}, \tag{10}
$$

where $\alpha$, $\beta$, and $\gamma$ are the hyper-parameter to balance the four loss items. Jointly optimizing the network by Eq. (10), we could finally learn local-sensitive cross-modal representation and effectively mitigate the heterogeneous modality gap. Our method is plug-and-play and does not alter the architecture of the dual-encoder CLIP model. During inference, like the vanilla CLIP, we extract the representation of a given query (image or text) using the corresponding encoder, and find the best matching target in the database by comparing cosine distances between all combinations.

## 4 EXPERIMENTAL RESULTS

### 4.1 Experimental Setup

**Datasets & Evaluation Metrics.** We evaluate the proposed framework on the typical Flickr30K [53] and MS-COCO [5] datasets, where each image is annotated with 5 texts. Following the dataset split in [3, 27], the Flickr30K dataset contains $29,000$, $1,000$, and $1,014$ images for training, testing, and validation, respectively. The MS-COCO dataset contains $123,287$ images. Following [35], we use $113,287$ images for training, $5,000$ images for validation, and $5,000$ images for testing. As a common practice in information retrieval, we measure the performance by the Recall at K (R@K) and RSUM. The higher R@K indicates better performance.

**Implementation Details.** Our method is designed to be plug-and-play, meaning it can be easily applied to existing image-text retrieval models without changing their original architectures. To validate the improved performance of our method in cross-modal retrieval, we conduct main experiments based on the popular pre-trained dual-encoder framework CLIP, including both ViT-B/16 and ViT-L/14 configurations [38]. The diffusion model (in Sec. 3.3) pertains to DALL-E 2 [39]. During training, the parameters of the image encoder and text encoder in CLIP are updated by the original contrastive loss, as well as the proposed local-driven and generative-driven loss. The image size is standardized at $224 \times 224$, while the maximum length of the text token sequence is defined as 77. We fine-tune the model for 6 epochs with batch size of 32. Adam Optimizer is used as the training optimizer, with an initial learning rate of $1 \times 10^{-5}$, and the cosine learning rate decay is applied [23]. As the investigation in Sec. 4.3, the hyperparameters $K$, $M$, $\alpha$, $\beta$, and $\gamma$ are set to 20, 5, 1, 0.98, and 0.01, respectively. *More details of our model can be found in the Code.*

### 4.2 Comparison with State-of-the-art Methods

**Compared Methods.** To validate the effectiveness of our approach, we evaluate our method by comparing its performance with a number of state-of-the-art methods including *without pre-training* (i.e., SCAN [27], IMRAM [3], VSE [4], SGRAF [11], NAAF [56], CHAN [36], HREM [16], and NUIF [55]), *partial pre-training* (i.e., VSE [4], VSRN++ [30], MV-VSE [31], CHAN [36], HREM [16], and NUIF [55]), and *pre-training* (CLIP$_{Vit-B/16}$ and CLIP$_{Vit-L/14}$) [38] methods. Specifically, the methods *without pre-training* usually employ the Faster RCNN [41] and ResNet-101 as the image encoder to extract region visual features, and adopt the *BiGRU* as the text encoder to learn textual feature. The *partial pre-training* methods replace the text encoder in the *without pre-training* with a pre-trained *BERT* [10]. The *pre-training* models have been optimized with amounts of text-image pairs, and show promising alignment ability compared with the *without pre-training* and *partial pre-training*

methods. We adopt the typical dual-encoder work *CLIP* [38] as the baseline, and integrate the proposed modules into it. Note that, the proposed method is plug-and-play, which does not affect the architecture of the *CLIP* model.

**Results Analysis.** Table 1 shows the performance comparison with the state-of-the-art *without pre-training*, *partial pre-training*, and *pre-training* cross-modal retrieval models. From the table, we can draw the following conclusions: **(1)** Among all the methods, the approaches *without pre-training* usually achieve low retrieval results. Compared with these approaches, although the *partial pre-training* methods just replace the text encoder with the pre-trained BERT model [10], they achieve remarkable improvements across both datasets, which indicates that making reasonable use of external knowledge is beneficial to facilitate the comprehension of complex data. Furthermore, by substituting the image encoder of the *partial pre-training* methods with the pre-trained model, the retrieval results can experience further enhancement. **(2)** We apply the proposed framework to *partial pre-training* method VSE∞ [4] and *pre-training* method *CLIP* [38] (denoted as *+LG-MGC*). The results are annotated with the purple background, which reveal that the proposed method can improve the performance of all baseline models, and achieve new state-of-the-art results on almost all metrics. Specifically, on the Flickr30K test set, we increase the RSUM of VSE∞ [4], CLIP$_{Vit-B/16}$, and CLIP$_{Vit-L/14}$ by 3.2%, 10%, and 7.2%, respectively. On the MS-COCO test set, our method can also exceed the baselines with satisfactory improvements from 1.7% to 11.1% in terms of the RSUM. *Note that*, in the *CLIP+LG-MGC* method, consistent with the vanilla CLIP model, we just utilize the features derived from the class tokens for testing. The gains indicate that our method is capable of extracting local knowledge beneficial for fine-grained cross-modal retrieval and mitigating the modality gap between the image and text, proving the effectiveness of the proposed approach. **(3)** By integrating the retrieval results derived from global representations with the results based on local features, we observe further performance enhancements, as illustrated at the bottom of Table 1 marked with pink background. Additionally, we find that the ensemble results closely align with those from *CLIP+LG-MGC*, further demonstrating that the proposed modality gap correction module has already enhanced the CLIP model's ability to capture local information. Due to limited space, more quantitative results can be found in our supplementary material.

### 4.3 Ablation Studies

**Influence of Different Network Components.** To analyze our proposed method and show the benefits of each module, we design several variants of our approach. Specifically,

**(1) Baseline:** we adopt the typical CLIP$_{ViT-B/16}$ [38] as the *Baseline* model, and fine-tune it on the Flickr30K dataset.

**(2) +Local$_{exp}$:** this variant incorporates the explicit LSC module into the *Baseline* model. By comparing it with the *Baseline*, we can evaluate the effect of the proposed strategy that explicitly incorporates the local representation into the global features.

**(3) +Local$_{full}$:** this variant incorporates the implicit LSC module into the *+Local$_{exp}$*. Since both the explicit and implicit LSC modules do not add any trainable parameters to the *Baseline*, we can conveniently explore the influence of local information by comparing it with the *Baseline*. Furthermore, by comparing it with

**Table 1: Comparisons with state-of-the-art methods on Flickr30k and MSCOCO. † denotes the improved results by the authors compared to the original paper, while ‡ means ensemble results of two models.**

| Data Split → | Flickr30K (1K) | | | | | | | MS-COCO (5K) | | | | | | |
| Eval Task → | Image-to-Text | | | Text-to-Image | | | RSUM | Image-to-Text | | | Text-to-Image | | | RSUM |
| Method ↓ | R@1 | R@5 | R@10 | R@1 | R@5 | R@10 | | R@1 | R@5 | R@10 | R@1 | R@5 | R@10 | |
| *(Faster-RCNN, ResNet-101, BiGRU, without pre-training)* | | | | | | | | | | | | | | |
| SCAN$_{(ECCV'18)}$ [27] | 67.4 | 90.3 | 95.8 | 48.6 | 77.7 | 85.2 | 465.0 | 50.4 | 82.2 | 90.0 | 38.6 | 69.3 | 80.4 | 410.9 |
| IMRAM$_{(CVPR'20)}$ [3] | 74.1 | 93.0 | 96.6 | 53.9 | 79.4 | 87.2 | 484.2 | 53.7 | 83.2 | 91.0 | 39.7 | 69.1 | 79.8 | 416.5 |
| SGRAF$_{(AAAI'21)}$ [11] | 78.4 | 94.6 | 97.5 | 58.2 | 83.0 | 89.1 | 500.8 | 55.8 | 83.0 | 91.0 | 42.0 | 72.4 | 82.1 | 426.3 |
| VSE∞$_{(CVPR'21)}$ [4] | 76.5 | 94.2 | 97.7 | 56.4 | 83.4 | 89.9 | 498.1 | 56.6 | 83.6 | 91.4 | 39.3 | 69.9 | 81.1 | 421.9 |
| NAAF$_{(CVPR'22)}$ [56] | 81.9 | 96.1 | 98.3 | 61.0 | 85.3 | 90.6 | 513.2 | 58.9 | 85.2 | 92.0 | 42.5 | 70.9 | 81.4 | 430.9 |
| CHAN$_{(CVPR'23)}$ [36] | 79.7 | 94.5 | 97.3 | 60.2 | 85.3 | 90.7 | 507.8 | 60.2 | 85.9 | 92.4 | 41.7 | 71.5 | 81.7 | 433.4 |
| HREM$_{(CVPR'23)}$ [16] | 81.4 | 96.5 | 98.5 | 60.9 | 85.6 | 91.3 | 514.3 | 60.6 | 86.4 | 92.5 | 41.3 | 71.9 | 82.4 | 435.1 |
| NUIF$_{(AAAI'24)}$ [55] | 84.3 | 96.3 | 98.0 | 60.7 | 85.0 | 90.7 | 515.1 | 61.8 | 86.8 | 93.1 | 43.3 | 72.4 | 82.6 | 439.8 |
| *(Faster-RCNN, ResNet-101, BERT, partial pre-training)* | | | | | | | | | | | | | | |
| VSRN++$_{(TPAMI'22)}$ [30] | 79.2 | 94.6 | 97.5 | 60.6 | 85.6 | 91.4 | 508.9 | 54.7 | 82.9 | 90.9 | 42.0 | 72.2 | 82.7 | 425.4 |
| MV-VSE$_{(IJCAI'22)}$ [31] | 82.1 | 95.8 | 97.9 | 63.1 | 86.7 | 92.3 | 517.5 | 59.1 | 86.3 | 92.5 | 42.5 | 72.8 | 83.1 | 436.3 |
| CHAN$_{(CVPR'23)}$ [36] | 80.6 | 96.1 | 97.8 | 63.9 | 87.5 | 92.6 | 518.5 | 59.8 | 87.2 | 93.3 | 44.9 | 74.5 | 84.2 | 443.9 |
| HREM$_{(CVPR'23)}$ [16] | 84.0 | 96.1 | 98.6 | 64.4 | 88.0 | 93.7 | 524.2 | 64.0 | 88.5 | 93.7 | 45.4 | 75.1 | 84.3 | 450.9 |
| NUIF$_{(AAAI'24)}$ [55] | 85.6 | 97.2 | 98.6 | 69.8 | 90.4 | 94.4 | 535.9 | 67.8 | 89.8 | 94.8 | 49.9 | 77.9 | 86.7 | 439.8 |
| VSE†∞$_{(CVPR'21)}$ [4] | 80.5 | 96.1 | 98.0 | 61.3 | 85.9 | 91.5 | 513.3 | 59.1 | 85.1 | 92.2 | 42.4 | 73.0 | 83.0 | 434.8 |
| **+LG-MGC (Ours)** | 82.4 | 95.8 | 98.0 | 61.5 | 86.9 | 91.9 | **516.5** | 59.0 | 85.8 | 92.4 | 42.8 | 73.1 | 83.4 | **436.5** |
| *(Dual-Encoder, pre-traning)* | | | | | | | | | | | | | | |
| CLIP†$_{Vit-B/16}$ | 88.4 | 98.7 | 99.5 | 76.1 | 94.6 | 97.2 | 554.5 | 65.2 | 87.3 | 92.2 | 50.3 | 76.0 | 84.2 | 455.2 |
| **+LG-MGC (Ours)** | 92.6 | 99.5 | 99.7 | 78.9 | 95.5 | 98.2 | **564.5** | 67.6 | 88.5 | 93.8 | 51.2 | 77.9 | 86.4 | **465.3** |
| CLIP†$_{Vit-L/14}$ | 90.7 | 99.0 | 99.6 | 77.3 | 94.6 | 97.7 | 558.9 | 65.7 | 87.2 | 92.8 | 50.2 | 76.6 | 84.9 | 457.4 |
| **+LG-MGC (Ours)** | 92.4 | 99.2 | 99.6 | 80.3 | 96.2 | 98.4 | **566.1** | 66.3 | 87.7 | 93.4 | 51.6 | 77.2 | 85.7 | **461.9** |
| **CLIP‡$_{Vit-B/16}$ (Ensemble)** | 93.1 | 99.7 | 99.8 | 78.9 | 95.5 | 98.2 | **565.2** | 68.1 | 88.6 | 94.0 | 51.1 | 77.8 | 86.3 | **466.0** |
| **CLIP‡$_{Vit-L/14}$ (Ensemble)** | 92.7 | 99.2 | 99.6 | 80.3 | 96.3 | 98.3 | **566.4** | 67.3 | 87.6 | 93.8 | 51.6 | 77.1 | 85.7 | **463.1** |

the *+Local$_{exp}$*, we can evaluate the effect of the implicit LSC module on the retrieval task.

**(4) +G$_{train}$:** this variant adds the GST module into the *Baseline*, in which the generative model (i.e., the DALL-E 2 [39]) is trained with the *Baseline* in an end-to-end manner.

**(5) +G$_{fix}$:** this variant fixes the parameters of the generative model in *+G$_{train}$*, leaving the rest unchanged. By comparing it with the *Baseline*, the effectiveness of the proposed GST can be verified. Furthermore, by contrasting it with the *+G$_{train}$*, we can evaluate the influence of varying generative mechanisms.

**(6) +Local$_{full}$&G$_{fix}$ (Ours):** this variant integrates both the explicit and implicit LSC module along with the fixed GST module into the *Baseline*, thereby allowing us to verify the effectiveness of our proposed approach.

**Table 2: Ablation studies on Flickr30K dataset.**

| Eval Task → | Image-to-Text | | | Text-to-Image | | | RSUM |
| Method ↓ | R@1 | R@5 | R@10 | R@1 | R@5 | R@10 | |
| CLIP†$_{Vit-B/16}$ | 88.4 | 98.7 | 99.5 | 76.1 | 94.6 | 97.2 | 554.5 |
| +Local$_{exp}$ | 91.8 | 98.8 | 99.9 | 76.9 | 94.9 | 97.4 | 559.8 |
| +Local$_{all}$ | 91.9 | 99.1 | 99.7 | 77.6 | 95.6 | 98.0 | 561.9 |
| +G$_{train}$ | 88.2 | 98.2 | 99.8 | 75.9 | 94.3 | 97.7 | 552.9 |
| +G$_{fix}$ | 91.9 | 99.3 | 99.8 | 78.1 | 95.3 | 98.1 | 562.4 |
| +Local$_{all}$&G$_{fix}$ | 92.6 | 99.5 | 99.7 | 78.9 | 95.5 | 98.2 | 564.5 |

Table 2 shows the ablation study results. From the table, we can conclude the following observations: **(1)** as expected, among all the variants, the *Baseline* gets the weakest performance, and our method could improve the base model by a clear margin. By comparing it with the *+Local$_{exp}$*, *+Local$_{full}$*, and *G$_{fix}$*, we can infer that the performance of the *Baseline* is constrained due to its inadequate depiction of the fine-grained information and the inefficient ability in mitigating the modality gap. **(2)** The results from *+Local$_{exp}$* and *+Local$_{full}$* indicate that capturing the local information, especially in the explicit manner, is critical for the fine-grained image-text

retrieval task. **(3)** Compared with the *Baseline* and $G_{fix}$, the variant *+G$_{train}$* demonstrates a clear decline in terms of the RSUM. We speculate that this phenomenon primarily arises from the necessity of sufficient paired data to fully train the generative model. Otherwise, fine-tuning the model without extra design may lead to overfitting, thereby affecting the performance. **(4)** The results shown in the last line in Table 2 are from our full model. As can be seen, it consistently outperforms other incomplete solutions, which indicates that both the LSC and GST module help improve the alignment of cross-modal embeddings.

**Influence of the Size of Local Information.** We investigate the influence of two main parameters involved in our proposed local-driven semantic completion strategy: the number of local region features $K$ and the response $M$ that integrated into the global feature in Sec. 3.2. Specifically, we train models for $K \in \{5, 10, 20, 30, 40, 50\}$ and $M \in \{1, 3, 5, 7, 9, 11\}$, and the results are depicted in Fig. 3. From Fig. 3 (a), we can observe that with more local information incorporated (5→20), better retrieval results can be obtained, and the performance converges at $K = 20$. The results in Fig. 3 (b) suggest that with the increase of the local response (1→5), the performance raises accordingly. When more local information is added, the final performance exhibits a certain degree of decline. Therefore, we set $K = 20$ and $M = 5$ in our experiments.

**Influence of Different Hyperparameters.** Fig. 4 shows the effect of the tradeoff parameters $\alpha$, $\beta$, and $\gamma$ in Eq. (10). Specifically, we vary $\alpha$ from 0.7 to 1.2 (in increments of 0.1), vary $\beta$ from 0.94 to 1.00 (in increments of 0.01), and vary $\gamma$ from 0.005 to 0.025 (in increments of 0.005) to control the weight of the explicit and implicit local-driven losses, as well as the generative-driven loss. From Fig. 4 (a), we find that the model performance increases with the increment of $\alpha$ and the optimal value is 1.0, while the performance decreases when $\alpha$ goes beyond the optimal value. Additionally, from Fig. 4 (b) and (c), we can observe that the optimal value of the $\beta$ and $\gamma$

are 0.98, and 0.01, respectively. We finally set $\alpha = 1.0$, $\beta = 0.98$ and $\gamma = 0.01$ in our experiments.

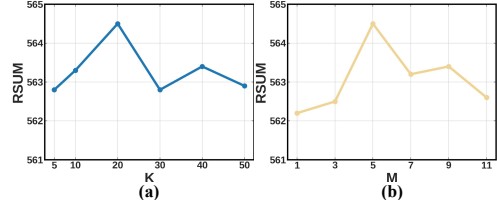

**Figure 3: Performance variation with respect to different sizes of local information, $K$ and $M$, on Flickr30K dataset.**

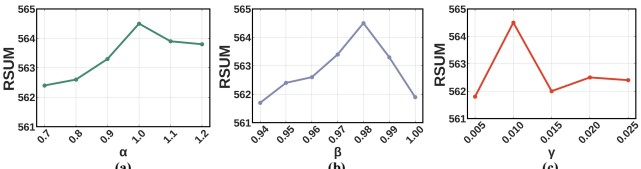

**Figure 4: Performance variation with respect to different parameters $\alpha$, $\beta$, and $\gamma$ on Flickr30K dataset.**

## 4.4 Visualization

**Visualization of the Patch-Wise Similarity Maps.** To qualitatively verify whether the proposed method can effectively incorporate fine-grained local features into the global embeddings, following [52], we calculate the similarity between each image patch and the global textual feature, which could be interpreted as the contribution of each patch feature to the global feature. For each sample, we visualized the similarity maps based on both the CLIP$_{Vit-B/16}$ and our model, and the results are shown in Fig. 5. Note that, in the figure, a brighter color indicates higher similarity, and a darker color means lower similarity. Specifically, from the qualitative results between each patch and the global feature (the first column of Fig. 5), we can observe that our method can precisely capture both objects (e.g., *man and women*) and environment information, while the CLIP$_{Vit-B/16}$ tends to focus on the dominated scenes. Correspondingly, compared with the CLIP$_{Vit-B/16}$, our method can obtain a higher similarity (indicated by the blue rectangular) between the image and text. Furthermore, when we replace the word *man* in the text with *girl*, *boy*, and *panda* (in the middle three columns), the proposed method shifts its focus away from the image areas that contain *man*. In contrast, the results from CLIP$_{Vit-B/16}$ show almost no change compared to the similarity with the original text, indicating its insensitivity to fine-grained local information. Additionally, when we replace the word *lake* with *grassland* (in the last column), the CLIP model completely ignores all areas of the image, whereas our method still accurately focuses on objects related to the unchanged words, such as *men and women*. Due to limited space, more qualitative results and detailed analysis are reported in the supplementary material.

**Visualization of the Modality Gap.** To testify whether our method can alleviate the heterogeneous modality gap, we compute the cross-modal distance and visualize it following the recipe from [32] in Fig. 6. Specifically, given 1000 image-text pairs from the test set of Flickr30K dataset, we first calculate the similarity between different samples based on euclidean distance (i.e., the

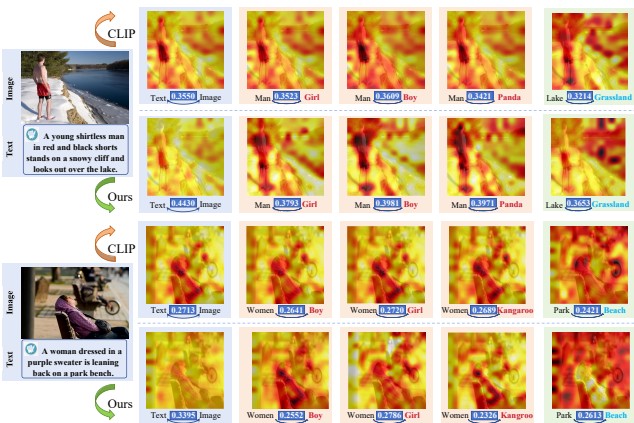

**Figure 5: Similarity maps from the vanilla CLIP and our proposed model on Flickr30K test set. The brighter the color, the higher the value.**

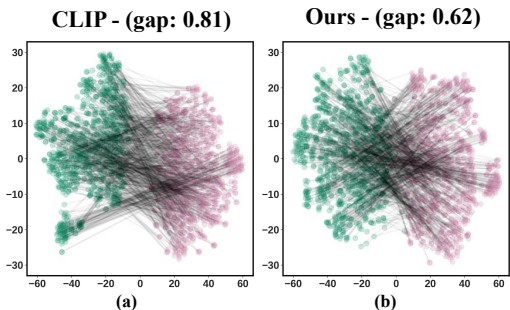

**Figure 6: Visualization of the modality gap for the CLIP (a) and our proposed method (b).**

distance between the blue dots and orange dots in Fig. 6). Moreover, the modality gap (shown in the top of Fig. 6) is assessed as the the difference between the center of image embeddings and text embeddings by $\Delta_{gap} = \frac{1}{n} \sum_{i=1}^{n} f_{cls_i}^v - \frac{1}{n} \sum_{i=1}^{n} f_{eos_i}^t$. From the comparison between Figure 6 (a) and (b), we can observe that the proposed model could clearly reduce the gap between different modalities. A possible explanation for this behavior could be that mapping textual features directly into the visual feature space can more effectively reduce the distance between them.

## 5 CONCLUSIONS

In this paper, to learn fine-grained semantic information and establish robust correspondence between image and text, we design a plug-and-play approach called LG-MGC. Our proposed model comprises two main components: a LSC module that supplements specific local context information within global representations, and a GST module that leverages the superior generative capabilities of a fixed diffusion model to translate textual features into the visual domain to enhance semantic flow. Through the innovative integration of the LSC and GST, our proposed model significantly enhances the performance of existing cross-modal retrieval models without adding extra trainable parameters, paving the way for a more intuitive and effective retrieval process. Extensive qualitative and quantitative experiments demonstrate the effectiveness of our proposed LG-MGC, achieving consistent state-of-the-art performances against strong baselines.

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
