# OpenReview forum: "Overcoming the Pitfalls of Vision-Language Model for Image-Text Retrieval"
_acmmm.org/ACMMM/2024/Conference — MM2024 Poster_

### Official Review · Reviewer_4xiw · 2024-05-08

**Rating:** 1
**Confidence:** 4

**Summary:**

The paper deals with the image-text retrieval task. The authors propose a local-driven semantic completion module and a generative-driven semantic translation module. Experiments prove the effectiveness of the proposed method.

**Strengths:**

The significant performance improvement.

**Limitations:**

1. The significant deviation from the prescribed template is problematic. The margins for the section titles are too small, which makes the document appear cluttered and unprofessional. It is crucial to adhere to the formatting guidelines to ensure readability and a polished presentation.

2. The overall quality of writing is subpar and difficult to follow. For instance, Figure 1 is not clearly explained, making it challenging for the reader to understand its relevance and implications. Additionally, the text within each figure is too small, hindering readability. Improving the clarity of explanations and ensuring that figure texts are legible will enhance comprehension.

3. The paper’s novelty is questionable. Figure 2 consists of numerous well-known models, and the authors do not introduce any new or surprising elements. To strengthen the paper, the authors should highlight unique contributions and novel insights that differentiate their work from existing models.

4. The use of numerous red and yellow figures is overwhelming for readers and detracts from the intended message. Such a vibrant color scheme can be distracting and may not effectively convey the information. Opting for a more subdued and coherent color palette will improve the visual appeal and ensure that the information is communicated more clearly.

Note: this review was written by the reviewer and polished by ChatGPT only for English usage. Your paper is not leaked to AI, and the opinions are from a real person.

**Suitability:**

3

---

### Official Review · Reviewer_VFwN · 2024-05-16

**Rating:** 4
**Confidence:** 3

**Summary:**

This paper found shortcomings in existing methods in bridging the fine-grained semantic gap between visual and textual representations, especially in capturing subtle interactions between local visual details and textual descriptions. Therefore, it proposes a general framework called Local and Generative-driven Modality Gap Correction (LG-MGC) to address the issues in the existing field of image-text retrieval.

**Strengths:**

1. The experimental design is rigorous, and the proposed methods have a certain level of innovation.

2. The research problem angle is quite novel, utilizing a local-driven semantic completion module to supplement traditional models by addressing specific local context situations that are ignored in global features. It uses generated features to alleviate modality differences, and is plug-and-play.

**Limitations:**

1. The two challenges described in the second paragraph of the Introduction seem to be related to differences in different modalities. What are the differences between them?

2. What does "† denotes the improved results by the authors compared to the original paper" mean in Table1; is there a problem with the content in Table2, $Local_{all}$ is not mentioned in the article, does it refer to $Local_{full}$?

3. Did the experimental results achieve performance improvement because both your image and text encoders used a clip with a large parameter size? Why did you choose DALL-E 2 as the generative model instead of other models?

**Suitability:**

3

---

### Official Review · Reviewer_UR1v · 2024-05-24

**Rating:** 2
**Confidence:** 3

**Summary:**

This paper introduces Local and Generative-driven Modality Gap Correction (LG-MGC) to enhance image-text retrieval models. LG-MGC comprises two modules: Local-driven Semantic Completion (LSC) and Generative-driven Semantic Translation (GST), addressing nuanced semantic gaps and modality discrepancies without added parameters. It's a plug-and-play solution, adaptable to existing models.

**Strengths:**

+ revealing the limitations of existing embedding-based approaches and proposing LG-MGC as a plug-and-play enhancement module,
+ introducing LSC and GST techniques to capture fine-grained cross-modal information effectively

**Limitations:**

- Lack of substantial novelty. Local and Global modeling has been extensively studied in prior work [a, b, c].

- Fig. 1 is interesting and impressive, but does the phenomenon exist extensively instead of limited data samples? More evidence should be provided.

- Unclear motivation for the Generative-driven Semantic Translation module. Why to adopt a diffusion model to enhance image-text retrieval? Besides, the usage of diffusion models may bring extra cost, making the retrieval system less efficient.

- The performance comparison between the proposed method and CLIP seems unfair, because the proposed method has been trained in Flickr30K and MS-COCO, while CLIP has not (i.e., the zero-shot setting). Fine-tuned variants of CLIP should be introduced to be compared.

- The generalization is limited, since the method is only applied to VSE\infinity and CLIP. More base models should be considered, such as more CLIP variants (e.g., CLIP-H, CLIP-G), ALBEF, ALIGN, and BLIP-2.

- The effectiveness is questionable. As shown in Tab. 1, the improvement over VSE\infinity is very marginal and those over CLIPs seems not significant either.

- In line. 905, the authors claim the gap is measured by the distance between two center points of different modalities, but it seems to calculate the mean of pairwise distance between any cross-modal points in Fig. 6, which is inconsistent.

Minors:
- The font size in Fig. 2 is too small. The whole figure is too abstract to be followed.
- There is not any blue or orange dots in Fig. 6 but described in line. 905.

[a] Diao et al. Similarity Reasoning and Filtration for Image-Text Matching. AAAI’2021

[b] Ji et al. Step-Wise Hierarchical Alignment Network for Image-Text Matching. IJCAI’2021.

[c] Ge et al. Cross-modal Semantic Enhanced Interaction for Image-Sentence Retrieval. WACV’2023

**Suitability:**

3

---

### Official Review · Reviewer_2dLW · 2024-05-29

**Rating:** 5
**Confidence:** 2

**Summary:**

The paper tackles the challenge of establishing stronger connections between image and text modalities by devising a model consisting of two primary parts: a local-driven semantic completion component that augments global features by incorporating crucial local context information previously disregarded by conventional models, and a generative-driven semantic translation module that utilizes generated features to reduce the disparity between the two modalities.

**Strengths:**

- I sympathize with the authors' claim: I believe that relevant nuances about the input are lost and cannot be recovered in the embedding space. This can be an issue when a stronger connection between text and image is needed.
- I like the authors' idea, and I think it is compelling and can be a starting point for future analyses

**Limitations:**

- Fig. should be revised to improve its clarity
- I believe that the authors should include details about the dimensionality of the variables involved in the equations they describe in sections 3.2 and 3.3, I think that their analysis would be much more significant to the reader
- I believe that some qualitative examples of what the authors claim in section 3.2 could help the reader understand their point
- I suggest the authors proofreading their manuscript for typos, and restructuring some sentences for clarity ("In this paper, to learn fine-grained semantic information and establish robust correspondence between image and text, we design a plug-and-play approach called LG-MGC." should be: "In this paper, we design a plug-and-play approach called LG-MGC to learn fine-grained semantic information and establish robust correspondence between image and text.")

**Suitability:**

3

---

### Meta-Review · Area_Chair_9GhR · 2024-06-30

**Recommendation:** Accept (Poster)
**Confidence:** 4

**Metareview:**

The paper initially received mixed ratings (WA, BA, WR, R) and after the rebuttal phase, even if two of the reviewers have raised or lowered their scores, the final scores have remained very mixed (WA, BA, WR, R). The AC has carefully read the paper, the reviewers and the response from the authors. Especially, the AC notes that some of the arguments brought by 4xiw (R) are not a sufficient basis for rejection (e.g. use of negative vspaces, use of vibrant colors in figures, small text inside figures, claim of a lack novelty because of the presence of existing components). Overall, considering the quality of the arguments different reviews, and the arguments raised towards acceptance or rejection, the AC finds the arguments towards acceptance to be more solid than those against acceptance. Therefore, the AC suggests accepting the paper as a poster.